
**Real-time probabilistic seismic hazard assessment based on seismicity anomaly**
Yu-Sheng Sun[1*], Hsien-Chi Li[1], Ling-Yun Chang[1], Zheng-Kai Ye[1], and Chien-Chih Chen[1, 2]
[1]Department of Earth Sciences, National Central University, Taoyuan City 32001, Taiwan, R.O.C.
[2]Earthquake-Disaster and Risk Evaluation and Management Center, National Central University,
Taoyuan City 32001, Taiwan, R.O.C.
Correspondence: Yu-Sheng Sun (sheng6010@gmail.com)
**Abstract**
The real-time Probabilistic Seismic Hazard Assessment (PSHA) is developed for considering the
practicability for daily life and the rate of seismic activity with time. The real-time PSHA follows
the traditional PSHA framework, but the statistic occurrence rate is substituted by time-dependent
seismic source probability. Pattern Informatics method (PI) is a proper time-dependent probability
model of seismic source, which have been developed over a decade. Therefore, in this research,
we chose the PI method as the function of time-dependent seismic source probability and selected
two big earthquakes in Taiwan, the 2016/02/05, Meinong earthquake ($M_L$ 6.6) and the 2018/02/06,
Hualien earthquake ($M_L$ 6.2), as examples for the real-time PSHA. The forecasting seismic
intensity maps produced by the real-time PSHA present the maximum seismic intensity for the



next 90 days. Compared to real ground motion data from the P-alert network, these forecasting
seismic intensity maps have considerable effectiveness in forecasting. It indicates that the real-
time PSHA is practicable and can provide a useful information for the prevention of earthquake
disasters.
**1   Introduction**
At present, there are two major phases about the researches and applications of seismic hazard: the
pre-earthquake and the post-earthquake. The most important usage of the post-earthquake seismic
hazard assessment is the Earthquake Early Warning (EEW) system (Cooper, 1868; Wu et al., 1998;
Wu et al., 2013). It provides extra time for people to take refuge before the larger seismic wave
arrives. On the other hand, Probabilistic Seismic Hazard Analysis (PSHA; Cornell, 1968; SSHAC,
1997) is the most common methodology of the pre-earthquake seismic hazard assessment and
mainly for engineering design. PSHA determines the exceeding probability of ground motion level
over a specified time period based on the occurrence rate of earthquake and ground motion
prediction equations (GMPEs). The occurrence rate of earthquake is generally described by the
truncated exponential model (Cosentino et al., 1977) and the characteristic earthquake model
(Schwartz and Coppersmith, 1984; Wang et al., 2016). No matter the data is from long-term
observations or paleoseismic studies, the earthquake occurrence rate computed from these models



will not change with time. However, the seismic activity is a complex dynamic process in time and
space and usually fluctuates enormously in short time scale (Chen et al., 2006). Furthermore, the
assessment is usually computed by using very long recurrence interval, 475 or 2475 years, for the
purpose of engineering design (Iervolino et al., 2011). As a result, it is hard to verify the accuracy
of seismic hazard assessment in limited life because of such long period. On the other hand, such
long interval is suitable for buildings, but not for human's life which is definitely much shorter
than the life span of buildings. In other words, the concept of catastrophic in such long recurrence
intervals is difficult to resonate in the daily life of general public. In addition, the definition like
10% probability in 50 years is hard to image for most ordinary people. Therefore, a statistical long-
term seismic hazard assessment is useless in our daily life. On the contrary, we believe that a short-
term and time-dependent pre-earthquake hazard assessment is necessary for everyone's daily use.
In this study, we suggested a preliminary method to achieve this goal by using a time-dependent
seismic source probability instead of the static one in the long-term assessment. One of the capable
candidates as a time-dependent seismic source probability is the Pattern Informatics (PI) method,
which has developed over the past decade (Rundle et al., 2000; Tiampo et al., 2002; Wu et al.,
2008a; Chang et al., 2016).

Anomalous change in seismicity is widely used as precursory indicator for big earthquakes and is



usually classified into seismic activation or seismic quiescence, depending on ascending or
descending number or occurrence rate of seismicity (Chen et al., 2005; Wu et al., 2008b). In the
PI method, big earthquakes tend to occur after precursory anomalous seismic changes and its
occurrence probability can be quantified by the magnitude of spatiotemporal variation of
seismicity. In preliminary researches, PI performs good in identifying locations nearby upcoming
big earthquakes. A modified version of PI developed in the recent researches has obviously
improved the accuracy of identifying occurrence time interval of big earthquakes. The occurrence
probability of big earthquakes in the next 90 days is plausible after a series of verification (Chang
et al., 2016; Chang, 2018). Therefore, we used the modified PI method to compute the time-
dependent seismic source probability of each location while the area of interest is coarse-grained
by square in uniform size.

In this research, we illustrate a simple way to achieve a real-time seismic hazard assessment. The
crucial step is to replace statistical seismic probability by the time-dependent probability from the
modified PI method. The real-time seismic hazard assessment produced the seismic hazard
forecasting maps for the next 90 days. The "real-time" PSHA can be updated with earthquake
catalog refreshing (time-dependent) and forecast for the near future (short-term), and compared
with the forecasting time scale and static seismic rate of the traditional PSHA, these can be called



"real-time". We illustrated this real-time assessment process by two recent big earthquakes in
Taiwan, the 2016 Meinong earthquake ($M_L$ 6.6) (Lee et al., 2016; Chen et al., 2017; Lee et al.,
2017) and the 2018 Hualien earthquake ($M_L$ 6.2) (Hsu et al., 2018). Detailed parameters about
these two earthquakes are listed in Table 1. Finally, the reliability of the seismic hazard maps was
verified by comparing with real ground motion data recorded by the P-alert network.

**2    Data**
**2.1 Central Weather Bureau Seismic Network (CWBSN) catalog**
We used the CWBSN catalog maintained by the Central Weather Bureau (CWB), Taiwan
(https://www.cwb.gov.tw/V7e/earthquake/seismic.htm    and    http://gdms.cwb.gov.tw/index.php,
last access: July 2018). The completeness magnitude ($M_c$) of this catalog is estimated
approximately 2.0 in local magnitude ($M_L$) (Wu et al., 2008c). In the analysis of focal depth, Wu
et al. (2008b) showed that the focal depth for about 80% earthquakes is shallower than 30 km.
Therefore, we used $M_L$ 2.0 and 30 km as the threshold of magnitude and focal depth to select
earthquakes used in the PI calculation.

**2.2 P-alert network**
In this research, the ground motion recordings from the P-alert network were used to verify the



effectiveness of the real-time seismic hazard assessments from our idea. The EEW research group
of the National Taiwan University (NTU) have begun to set up the P-alert real-time strong-motion
network since 2010. The device of the P-alert network can record real-time acceleration signals in
three-component and publish alerts if the peak initial-displacement amplitude (*Pd*) or peak ground
acceleration (PGA) exceeds a redefined threshold (Wu et al., 2013, 2016b). Nowadays, there are
more than 600 stations in Taiwan; most of them are located in elementary schools (Wu et al., 2013;
Yang et al., 2018). We mainly adopted the P-alert waveform database maintained by Taiwan
Earthquake research Center (TEC) and the data from NTU were as an auxiliary catalog (The data
of the P-alert network can be downloaded from the Data Center of TEC:
http://palert.earth.sinica.edu.tw/db/ (last access: July 2018) or contact with Prof. Yih-Min Wu at
NTU for NTU's catalog: drymwu@ntu.edu.tw). However, even if there are so many stations
covering Taiwan, the distribution of the P-alert network is still nonuniform (see Fig. 2b and 3b).
This nonuniform distribution may causes some problems that we will discuss later.

**3    Method**
**3.1  Pattern Informatics (PI)**
The physical fundamental of the PI method is phase dynamics which describes changes of a system
by rotation of state vector in the Hilbert space (Rundle et al., 2002; 2003). The evolution of state


vector in a dynamic fault system is suggested to be related to stress accumulation and release (Chen
et al., 2006). The computation steps we addressed here are a modified version developed by Chang
et al. (2016) and Chang (2018) to improve temporal resolution of PI. The research area (119°~123°
E 21°~26° N) is divided into boxes of grid size 0.1°×0.1°, and each box is indicated by parameter
$x_i$. Because of the $M_c$ and the distribution of focal depth (mentioned in Section 2.1), all events
having $M_L \geq 2.0$ and depth $\leq 30$ km were used. In the PI computation, $t_1$ and $t_2$ are the beginning
and end of a change interval, respectively, and the length of change interval is 4 years. The
beginning time of calculation, $t_0$, is defined as 12 years before $t_2$. Then, $t_b$ is a sampling
reference time between $t_0$ and $t_1$ which shifts $t_b$ each time. The forecasting interval, $t_3$, starts
after $t_2$. According to Chang et al. (2016), the forecasting interval of the PI method reaches 90
days. Lastly, the PI method produces a forecasting probability distribution of seismic sources for
$M_L \geq 5.0$ within the forecasting interval.

**3.2  Real-time PSHA**
In the traditional PSHA framework (Cornell, 1968; Wang et al., 2016), the probability of an
earthquake's occurrence follows the Poisson process and the average recurrence interval for an
annual frequency of exceedance can be expressed as




$$v(Z > z) = \sum_{i=1}^{N_S} \dot{N}_i \iint f_{M_i}(m)\, f_{R_i}(r)\, P(Z > z \mid m, r)\, dm\, dr$$

127                                                                                                (1)

where $f_{M_i}(m)$ and $f_{R_i}(r)$ are the probability density functions of magnitude and distance,
respectively; $P(Z > z \mid m, r)$ is the conditional probability of ground motion $Z$ exceeding a
specified value $z$ for a specific magnitude $m$ and distance $r$. $\dot{N}_i$ is the annual occurrence rate
of earthquakes and described by the truncated exponential model (Cosentino et al., 1977) and the
characteristic earthquake model (Schwartz and Coppersmith, 1984). Finally, to consider all
scenarios, the total probability of $N_s$ earthquakes is summarized in a given region.

In the real-time PSHA, the occurrence rate of earthquake used in the traditional PSHA framework
is replaced by seismic forecasting probability to achieve spatiotemporal variability of the hazard
assessment. Then, considering the gridded space, the real-time PSHA can be expressed as
$$v(Z > z) = \sum^{M_S} \sum^{Loc_S} P_{M_i, Loc_i}(m, loc)\, P(Z > z \mid m, loc)$$

139                                                                                                (2)

where $P_{M_i, Loc_i}(m, loc)$, the forecasting probability distribution, is a function of magnitude and
location. It specifies an occurrence probability for specific magnitude, $M_i$, at each spatial location,
$Loc_i$. The summations are to consider the whole of the contribution from any possible magnitude,




$M_s$, and location, $Loc_s$. In this research, we adopted the forecasting probability from the PI method
as $P_{M,Loc}(m, loc)$. $Loc$ refers to $x_i$ in the PI method. The forecasting probability of the PI
method presents a distribution of cumulative forecasting probability for $M_L \geq 5.0$. Thus, we
referred to the average character of Gutenberg-Richter law in Taiwan (Gutenberg and Richter,
1944; Wang et al., 2015) to turn it into probability density function (PDF). It can be corresponded
to the specific magnitude conditions for $P(Z > z \mid m, loc)$. To evaluate the ground motion, we
used the GMPE published by Lin et al. (2012), which was also adopted for the issue of Taiwan
PSHA in Lee et al. (2017). In this GMPE, the earthquake type is one of the important parameters.
However, the divisions of seismic source in the PI method is no longer based on the geological
classification, but the grid box, $x_i$. Considering that the most faults in Taiwan are reverse faults
(Shyu et al., 2016), we adopted the reverse fault parameters setting for the entire research area.
Finally, the forecasting maximum PGA from the real-time PSHA is transferred to seismic intensity
according to the seismic intensity scale of CWB listed in Table 2 (Wu et al., 2003). It means that
the forecasting seismic intensity map presents the maximum seismic intensity which every site
will encounter in the following 90 days.

**3.3 Performance verification**
**3.3.1    Receiver Operating Characteristic curve (ROC)**



The ROC diagram is a binary classification model and widely used as a tool for quantifying the
performance of earthquake prediction (Holliday et al., 2006; Nanjo et al., 2006; Wu et al., 2016a).
We used the ROC diagram as an objective quantitative indicator to evaluate the performance of
the forecasting seismic probability computed by the PI method. For each box $x_i$, there are four
situations, parameters, while comparing forecasting hotspot and target earthquake: $a$ means any
target earthquake in a hotspot; $b$ means no target earthquake in a hotspot; $c$ means no hotspot
but with at least one target earthquake; $d$ means no target earthquake and no hotspot. True
positive rate (TPR) is defined as $a/(a + c)$ and false positive rate (FPR) is defined as $b/(b + d)$.
The values of $a$, $b$, $c$, and $d$ change with threshold of forecasting probability, and therefore
TPR and FPR change as well. The area under the ROC curve (AUC) is between 0 and 1. AUC=1
is a perfect prediction; AUC=0.5 is a random guess. For each forecasting map of PI, we generated
1000 random tests by re-distributing the hotspots randomly over the research area to examine the
possibility that a specific distribution of hotspots can generate by chance. In Fig. 1c and 1d, the
blue line is the 95% confidence interval based on two standard deviations. The standard deviation
is calculated by the random test results in each bin of the x-axis. The 95% confidence interval helps
us differentiate the distributing range of random tests and the significant of forecasting probability.

**3.3.2   Average Percent Hit Rate (APHR)**


The success rate of forecasting seismic intensity is a predictive accuracy of classification problems
for which the average percent hit rate (APHR) is arguably the most intuitive measure of
discrimination. The APHR is a rate at which the forecasting data are classified into the correct
classes (Sharda and Delen, 2006). In this research, the APHR was used to quantify the forecasting
performance of the real-time seismic hazard assessments. In the APHR, the exact hit rate which
only counts the correct classifications to the exact same class can be expressed as:
$$\text{APHR}_{\text{exact}} = \frac{1}{N} \sum_{i=1}^{g} p_i$$

186                                                                                                  (3)

where, in our case, $N$ is the total number of the P-alert stations or the boxes on the forecasting
hazard map, $g$ is the total number of seismic intensity classes (=8, according to the CWB's
seismic intensity scale), and $p_i$ is the total number of samples classified as class $i$. In the random
test, we further generated 1000 random tests by randomly re-distributing the forecasting maximum
seismic intensity over the research area and the stations to examine the possibility that a specific
distribution of the forecast can generate by chance.

**4    Results**
**4.1    Forecast of earthquake occurrences**



Figure 1a and 1b show the forecasting probability maps computed by the PI method, and Fig. 1c
and 1d are corresponding forecasting performance verified by the ROC tests. In the case of 2016
Meinong earthquake, $t_0$, $t_1$, and $t_2$ are 2004/01/31, 2012/01/31, and 2016/01/31. In the case of
2018 Hualien earthquake, $t_0$, $t_1$, and $t_2$ are 2006/01/31, 2014/01/31, and 2018/01/31. The
forecasting intervals of both cases are 90 days after $t_2$. Cyan star in Fig. 1a and 1b is the main
shock of 2016 Meinong and 2018 Hualian earthquakes, and the biggest earthquake in the
forecasting interval. Gray circles in Fig. 1a and 1b are the earthquakes with magnitude $M_L \geq 5.0$
in the forecasting interval, and more detailed information about these earthquakes can be obtained
in Table 1. A notable point is that both main shocks and most big earthquakes are located in or
very close to the hotspots. The performance of the PI forecasting probabilities seems to be good
simply by visual inspection.

In Fig. 1c and 1d, red curves are far above the blue curves (95% confidence interval). The AUCs
of red curves are 0.91 and 0.94, and are apparently larger than the AUCs of blue curves, which are
0.73 and 0.70. The ROC tests verified quantitatively that the performance of the PI forecasting
probability is significant, and these patterns are not just generated by random distribution of
hotspots by chance. Both distributions of hotspot are physically meaningful. Therefore, we can use
these probability maps as the function of earthquake occurrence rate in subsequent calculation for



the real-time PSHA.

**4.2  Real-time PSHA**
In Fig. 2 and 3, panel (a) shows the map of forecasting max seismic intensity estimated by the real-
time PSHA for the forecasting interval; panel (b) shows the map of max seismic intensity recorded
by the P-alert network during the forecasting interval. To ensure that it is absolutely maximum
intensity during the forecasting interval, we only used the stations which have recorded all the
target events ($M_L \geq 5.0$) in the forecasting interval. Although there are over 600 P-alert stations
distributing widely in Taiwan, some boxes still do not contain any station, for example, the Central
Mountain Range (see Fig. 5a and 5b). Therefore, we had to estimate the intensities in such kind of
boxes by interpolating. Thus, this strategy indeed generates the artificial effect and we will show
it later.

Comparing Fig. 2a and 2b, we suggest that both seismic intensity distributions are very similar.
An apparent deviation of forecasting seismic intensities from the recorded values is in the
southwestern Taiwan, especially the area closer to the 2016 Meinong main shock. Fig. 2c shows
the difference of intensity between Fig. 2a and 2b; the color of blue and red means that the
forecasting  value  in  a  box  is  underestimated  or  overestimated.  Most  boxes  have  intensity



difference in the range -1 to 1, but some boxes in the southwestern Taiwan are underestimated; the
differences are most 2 or even up to 3.

Comparing Fig. 3a and 3b, we suggest that both seismic intensity distributions are still very similar.
In this case, an apparent deviation of forecasting seismic intensities from the recorded values is in
the southern Taiwan and a part of southwestern area. Figure 3c shows that most boxes in the
southern Taiwan have smaller recorded intensity, and the recorded intensities in a part of
southwestern Taiwan are larger than the forecasting values.

Figure 4 shows the verifications generated by the APHR to quantitatively evaluate the performance
of the forecasting seismic intensity. We considered the denominator of two classifications in Eq.
3, i.e. the total number of the P-alert stations and the total number of boxes in the research area.
The results are indicated by "P-alert" and "Map" in Fig. 4, respectively. While comparing
forecasting intensity to recorded value, both cases "forecasting = recorded" and "forecasting =
recorded +1" belong to "successful forecasting". The definition of the tolerance range that depends
on the perspectives and allowance of different users is certainly debatable (Hsu et al., 2018). In
our case, the reason is that considering to prevent or mitigate earthquake disaster, "overestimation"
is better than "underestimation". Therefore, we tolerated the case of overestimation of 1 intensity



rather than underestimation.

First, all red lines are above the maximum hit rate of random tests and higher than 0.5, not to
mention the random guess of the eight choices of the seismic intensity scale. It means that these
forecasting seismic intensity maps have considerable effectiveness in the forecast, and their good
performance can't merely happen by chance. Moreover, another property is that both hit rates of
the "P-alert" cases are higher than the rates of the "map" cases. This result could be attributed to
the influence of the artificial effect generated by the interpolation of seismic intensity from the P-
alert data of nonuniform distribution. Last, it is emphasized that we just focus on the earthquakes
with $M_L \geq 5$ in this research, but we cannot deny the possibility of a $M_L < 5$ earthquake to cause
large seismic intensity in the near field.

**5    Discussion**
The results of the APHR performance test indicates that the maps and stations of forecasting max
seismic intensity by the real-time PSHA are significant and effective. Figure 5 is a concretization
of the APHR verification and further gives more details. It clearly shows the P-alert station
distributions of the "hit" and "not hit", considering only the station-to-station prediction
relationship between the forecasts and records. In both cases, most of the P-alert stations are hit



268 (Fig. 5a and 5b), and the hit percentages distribute along the diagonal and tolerant ranges (Fig. 5c

269 and 5d). However, there still are some locations or stations with wrong forecast. In the case of

270 2016 Meinong earthquake, the stations located in the southwestern Taiwan do not match the real

271 records, and at high seismic intensities (>3), the forecasting results at some stations are

272 underestimated (Fig. 5c), especially in the southwestern area. In the case of 2018 Hualien

273 earthquake, the result from the "P-alert" APHR seems better than former, and further the

274 distribution of the hit percentage is more concentrated along the diagonal and tolerant ranges.

275 Nevertheless, the stations in the southern and part of southwestern Taiwan are still missed. These

276 abovementioned differences between forecasting results and recorded seismic intensities in both

277 cases can be mainly attributed to three aspects.

278

279 First of all, the forecasting model that determines the probability distributions of earthquake

280 occurrences is critical for the real-time PSHA. If the probability distribution is missing or false

281 alarm in somewhere, it directly causes the inaccurate forecasts to the real-time PSHA. In the PI

282 results, some differences are located on the hotspots with relatively higher probability, for example,

283 the area in 22.6º to 23ºN and 120.9º to 121.3ºE in Fig. 1a, and 22.7º to 23.1ºN and 120.4º to 120.8ºE

284 in Fig. 1b. Compared the locations of the earthquakes, these hotspots just shift slightly and it seems

285 acceptable. However, in the results of the real-time PSHA, it leads the maps of forecasting max


seismic intensity to underestimate in the area near the epicenters and overestimate in the area
without any earthquake event, but with high probability of earthquake occurrence. For instance,
the southwestern area in the case of 2018 Hualien earthquake is underestimated because of this
reason, and then it also causes overestimated in the southern area (see Fig. 3 and 5b). Therefore, a
more accurate and precise forecasting model helps us get a more positive result in a real-time
PSHA. Even if the PI results perform well in the ROC test, the PI method still needs to be improved.

Secondly, the evaluation of earthquake ground motion suffers from the limitations of GMPEs. We
adopted the GMPE produced by Lin et al. (2012) whose data ($M_L \geq 5.0$) within 50 km are less than
14% of all data for the regression of GMPE. If there is a shortage of data in near field and for larger
events in the regression of GMPEs, the applicability of GMPEs is limited (Edwards and Fäh, 2014).
Therefore, that probably causes the deviation of evaluation on forecasting seismic intensity maps,
for instance, the underestimation of the areas around the two main shocks (Fig. 2c and 3c).
Moreover, the site effect is difficult to be properly and comprehensively evaluated in GMPEs, but
it dramatically affects the behavior of seismic waves. For example, the amplitudes in the Meinong
earthquake were amplified extending along the northwest (in Fig. 2b) because of the Western Plain
composed of thick and low velocity sedimentary deposits (see Fig. 4 in Lee et al., 2016). As a
result, the site effect also contributes and leads the seismic intensity forecast to underestimate (Fig.



2c and 5a).

Last but not least, the directivity effect also plays an important role in the distribution of ground
motion. For the main shocks in two cases, the rupture characteristic brings a strong directivity
effect that causes the significant amplification of ground motion along the rupture direction (Lee
et al., 2016; Hsu et al., 2018). However, GMPEs are basically a statistical distribution of PGA
generated by all data at the same radical distance without considering possible effect of rupture
directivity. As a result, GMPEs are only able to provide the ground motion estimation of radial
extension. Besides, the forecasting model does not include the information of rupture direction
either. Therefore, we suggest that some differences which along the rupture direction may belong
to this effect.

**6    Conclusion**
This study presents how the real-time seismic hazard assessment can be achieved by replacing the
static seismic rate, i.e. the truncated and characteristic earthquake models, with the time-dependent
seismic source probability of the PI method. With regard to the time-dependent seismic source
probability, the ROC tests verified quantitatively that the performances of the PI forecasting
probabilities in forecasting interval are quite effective. Therefore, those significant probability



distributions can be used as the function of earthquake occurrence rate, $P(m, loc)$, in the real-time
PSHA. Our forecasting seismic intensity maps of the real-time PSHA have the hit rates
outperformed the random guesses and higher than 0.5 for both cases of the Meinong and Hualien
earthquakes. This study thus suggests that these real-time PSHA maps are effective in terms of
forecasting, and their good performances are not likely coincidence. We demonstrated that the real-
time seismic hazard assessment is doable and can be realized and updated by the time-dependent
seismic source probability.

In the future, the different time-dependent seismic source probability models can be introduced to
provide a more accurate and robust estimation for earthquake occurrences. Also, a possible
improvement for our results could be from the estimated PGA distribution not only by means of
the state-of-the-art machine learning tools for a big data bank of the P-alert network but also by
physics-based numerical simulations (PBS) of seismic ground motion, instead of the empirical
GMPEs. Presumably, a real-time forecasting map of seismic intensity enables governments or
businesses to efficiently prepare for earthquake disasters. Moreover, the seismicity intensity scale
based on PGA are related to the vulnerability level of buildings, which will also be changed with
time due to the degradation and upgrades (e.g. obsolescence, retrofitting actions, climate events).
Therefore, when further assessing a seismic risk fluctuating with time, the real-time PSHA and the



change of vulnerability should be considered.

**Acknowledgments**
The authors are grateful for research support from the Ministry of Science and Technology (ROC)
and the Department of Earth Science, National Central University, Taiwan (ROC). This work is
supported by "Earthquake-Disaster & Risk Evaluation and Management Center, E-DREaM" from
The Featured Areas Research Center Program within the framework of the Higher Education
Sprout Project by the Ministry of Education (MOE) in Taiwan.

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

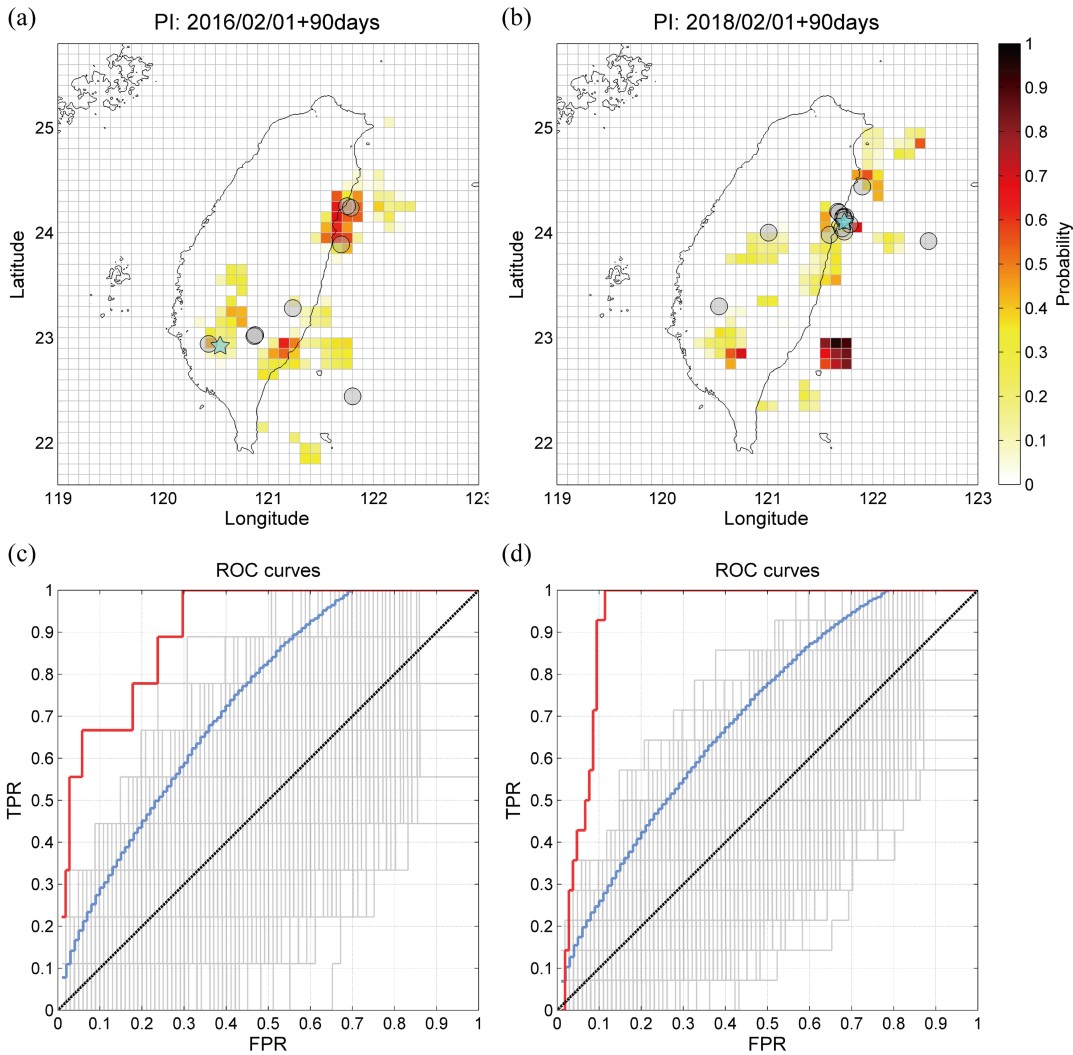


Figure 1. Panels (a) and (b) show the forecasting probability maps of the Meinong earthquake and

the Hualien earthquake, respectively. Panels (c) and (d) are the ROC curves of (a) and (b),

respectively. Red, gray, blue, and black curve represent the forecasting probability map, random

tests, 95% confidence interval, and the average of random tests, respectively.

458

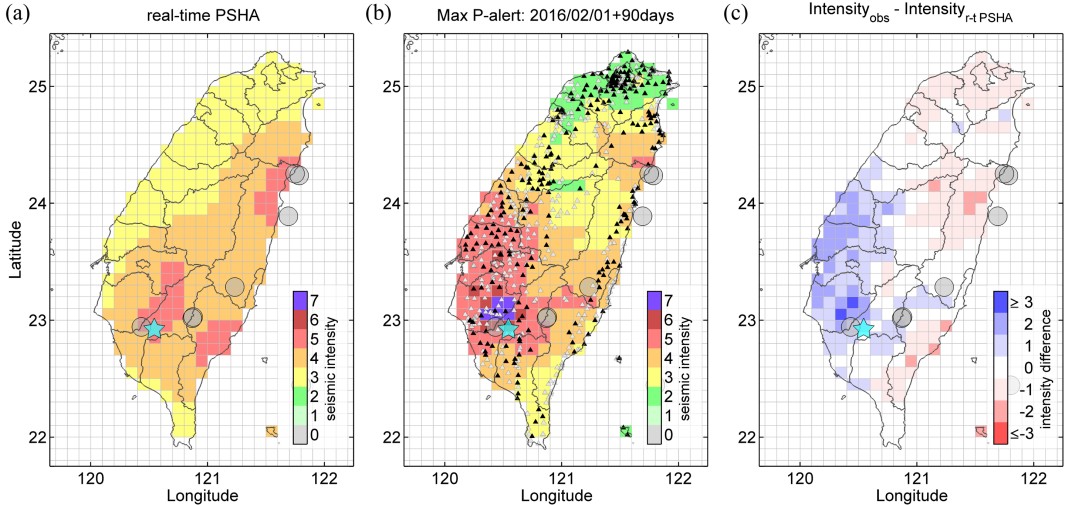

459

Figure 2. The case of 2016 Meinong earthquake: (a) The map of forecasting max seismic intensity

by the rea-time PSHA. The forecasting interval of seismic intensity is 90 days. (b) The map of

max seismic intensity recorded by the P-alert network. Black and white triangles indicate the P-

alert stations which we used and didn't use in the verification, respectively. (c) The difference of

seismic intensity between the forecast and the record. Cyan star represents the Meinong earthquake;

gray circles represent the earthquakes with magnitude $M_L \geq 5$ in this forecasting interval.






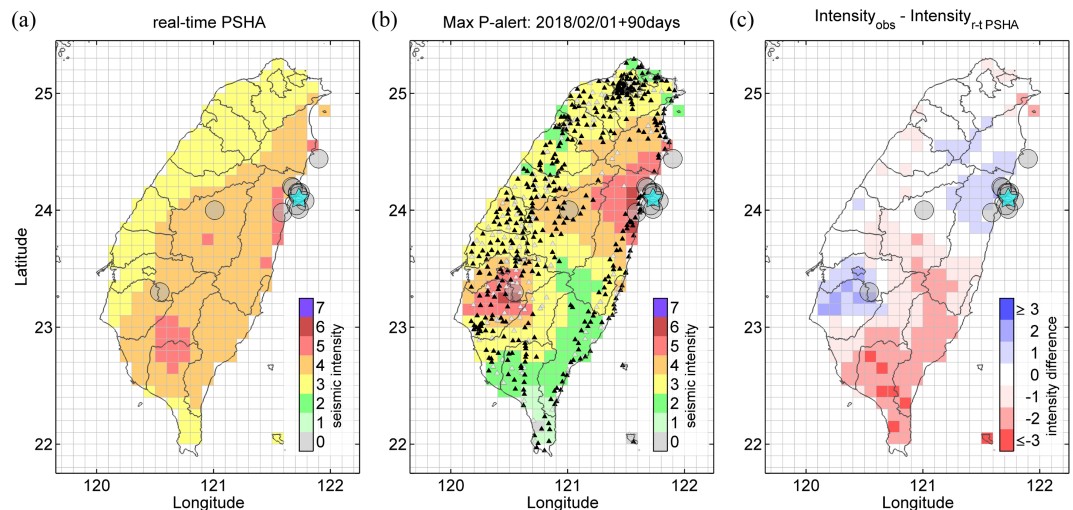


Figure 3. The case of 2018 Hualian earthquake: (a) The map of forecasting max seismic intensity.

(b) The map of max seismic intensity recorded by the P-alert network. (c) The difference of seismic
intensity between the forecast and the record. Cyan star represents the Hualian earthquake.






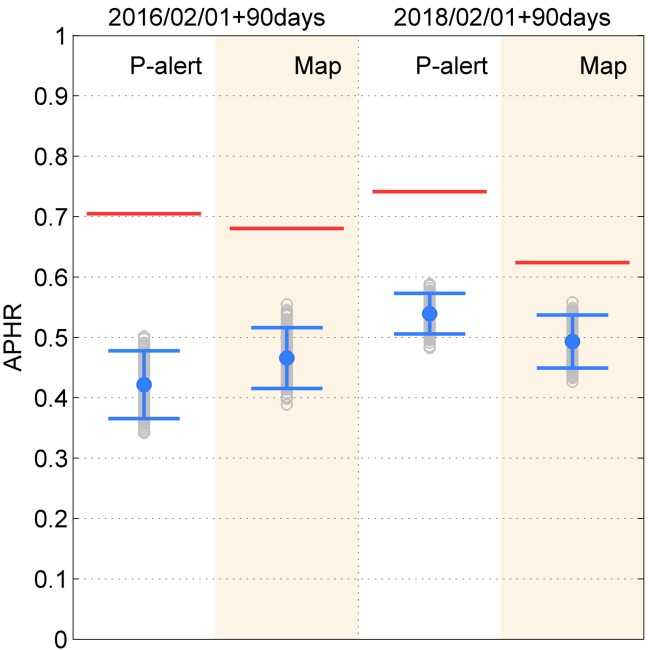


Figure 4. Performance test of APHR. Red line indicates the forecasts of the real-time PSHA; gray
circle indicates the result of a random test by randomly re-distributing seismic intensities; blue
error bar indicates the interval with two standard deviations over all random tests.






Figure 5. Panels (a) and (b) are the P-alert station distributions of the "hit" and "not hit". Red and

blue triangles present the "hit" and "not hit", respectively. Panels (c) and (d) are the distributions

of the hit percentage for the cases of 2016 Meinong and 2018 Hualian earthquake, respectively.



Red line area presents the acceptable prediction range.



Table 1. The earthquakes occurred in the forecast interval. "P-alert" indicates that the P-alert
recording obtained from the Taiwan Earthquake Research Center (TEC) or the National Taiwan
University (NTU). "Num." is the number of recording stations. "Nan" indicates that there is no P-
alert data to be recorded in both TEC and NTU even if that event was recorded by CWB. The bold
represents the Meinong and Hualian earthquakes.

(a) Meinong case: 2016/02/01~2016/05/01

| Date | Hour | Min. | Lon. | Lat. | Depth | $M_L$ | P-alert | Num. |
|---|---|---|---|---|---|---|---|---|
| **02/05** | **19** | **57** | **120.54** | **22.92** | **14.64** | **6.60** | **TEC** | **338** |
| 02/05 | 19 | 58 | 120.43 | 22.94 | 18.10 | 5.26 | Nan | Nan |
| 02/09 | 00 | 47 | 121.69 | 23.89 | 5.69 | 5.12 | TEC | 341 |
| 02/18 | 01 | 09 | 120.87 | 23.02 | 5.44 | 5.27 | TEC | 357 |
| 02/18 | 01 | 18 | 120.88 | 23.03 | 4.26 | 5.13 | TEC | 357 |
| 04/16 | 10 | 55 | 121.80 | 22.44 | 11.83 | 5.22 | TEC | 436 |
| 04/27 | 15 | 17 | 121.78 | 24.24 | 11.94 | 5.67 | NTU | 424 |
| 04/27 | 15 | 27 | 121.75 | 24.25 | 12.99 | 5.13 | NTU | 425 |
| 04/27 | 18 | 19 | 121.23 | 23.28 | 15.21 | 5.52 | NTU | 423 |


(b) Hualian case: 2018/02/01~2018/05/02

| Date | Hour | Min. | Lon. | Lat. | Depth | $M_L$ | P-alert | Num. |
|---|---|---|---|---|---|---|---|---|
| 02/04 | 13 | 12 | 121.67 | 24.20 | 15.10 | 5.10 | TEC | 543 |
| 02/04 | 13 | 56 | 121.74 | 24.15 | 10.60 | 5.80 | TEC | 519 |
| 02/04 | 13 | 57 | 121.68 | 24.19 | 11.10 | 5.10 | Nan | Nan |
| 02/04 | 14 | 13 | 121.72 | 24.15 | 10.30 | 5.50 | TEC | 517 |
| 02/05 | 15 | 58 | 121.72 | 24.14 | 10.00 | 5.00 | TEC | 522 |
| **02/06** | **15** | **50** | **121.73** | **24.10** | **6.30** | **6.20** | **TEC** | **520** |
| 02/06 | 15 | 53 | 121.59 | 23.98 | 5.10 | 5.00 | TEC | 520 |





| 02/06 | 18 | 00 | 121.73 | 24.12 | 6.70  | 5.30 | TEC | 516 |
|-------|----|----|--------|-------|-------|------|-----|-----|
| 02/06 | 18 | 07 | 121.71 | 24.04 | 4.20  | 5.30 | TEC | 516 |
| 02/06 | 19 | 15 | 121.73 | 24.01 | 5.70  | 5.40 | TEC | 516 |
| 02/07 | 15 | 21 | 121.78 | 24.08 | 7.80  | 5.80 | TEC | 523 |
| 02/25 | 18 | 28 | 121.90 | 24.44 | 17.70 | 5.20 | TEC | 533 |
| 03/20 | 09 | 22 | 120.54 | 23.30 | 11.20 | 5.30 | TEC | 539 |
| 03/29 | 00 | 17 | 121.01 | 24.00 | 11.10 | 5.00 | NTU | 388 |
| 04/23 | 17 | 10 | 122.53 | 23.92 | 19.30 | 5.10 | NTU | 381 |


Table 2. Seismic intensity scale of CWB.

| Intensity Scale | | Ground Acceleration (cm/s$^2$, gal) |
|-----------------|---|-------------------------------------|
| Micro           | 0 | <0.8     |
| Very minor      | 1 | 0.8~2.5  |
| Minor           | 2 | 2.5~8.0  |
| Light           | 3 | 8~25     |
| Moderate        | 4 | 25~80    |
| Strong          | 5 | 80~250   |
| Very Strong     | 6 | 250~400  |
| Great           | 7 | ≥400     |
