# Peer review of "Real-time probabilistic seismic hazard assessment based on seismicity anomaly"

_Natural Hazards and Earth System Sciences, 2019_

## Referee Comment (RC1) · Anonymous Referee #1 · 31 Jul 2019

The methods proposed here are well-considered and an excellent example of how time-dependent forecasting using seismicity-based tools. I do think that the PI method is not well enough described, and more details should be added on the estimation of ground motion using the GMPE method of Lin. Also, the English grammar needs considerable editing prior to publication - too much for reviewers at this stage.

---

## Referee Comment (RC2) · Anonymous Referee #2 · 6 Nov 2019

In this paper the authors develop a Real-time Probabilistic Seismic Hazard Assessment using a time dependent probability model (Pattern Informatics) as seismic source. For the calculation of the earthquake ground motion the authors use the attenuation laws produced and published by Lin et al. (2012). The methodology was applied in Taiwan, during the occurrence of two strong earthquakes in 2016 (Meinong) and 2018 (Hualien). In the first part of the paper the authors describe their approach on seismic hazard assessment and they explain, in a general way, the PI model used, the real-time PSHA and the procedures for verifying the performance of the results. In the second part the authors analyze the results obtained with the proposed methodology and compare them with the ground motion data recorded from the P-alert network. They conclude that, in future, the use of this method could be an important tool to esti-
mate PGA distribution to prepare us for earthquake disasters. The article is satisfactory and the methodological approach is described partially in the text, the other theoretical part about the PI model is delegated to other articles published by different research groups. I suggest that the paper must be publish with minor revision. I suggest only the following comments about the paper: 1) The paper, before publication, must be read by a native English speaker who will correct the English grammar; 2) It is important to describe the PI method so that the reader, who does not know the methodology, can understand the content of the paper; 3) The ROC diagram allows to guantify the goodness of one model with respect to another that is taken as a reference. In the case analyzed in the text the PI model gives a better performance than a Poissonian model and a random one. This result could be taken for granted because when using a seismicity information in a model, this leads to a better performance than very simple models. An interesting evaluation could instead be made using new test methods implemented for earthquake forecast models in the CSEP (Collaboratory for the Study of Earthquake Predictability) to understand the behavior of the model in space, number of earthquakes and so on. In Figure 1(b), for example, is evident a zone (9 cells) with the highest probability (equal to one) but with no associated hotspot. This area is not present in Figure 3, perhaps because, I think, it falls into the sea. How do you explain that these values do not affect the performance of your model? Why do you use only the ROC diagram and have not used others methods that provide important information about the performance of your model?

---

## Author Comment (AC2) · 4 Dec 2019

The authors would like to thank RC2 for the important suggestions, and we would add the PI description and do English editing before publication. We respond to the RC2's questions below.

1. The description of the PI method.

Please refer to AC1: 'reply RC1: 'Comments' and its Supplement.

2. How do you explain that these values (the zone (9 cells) with the highest probability) do not affect the performance of your model?

We separated the probability distribution of Hualien earthquake (Fig. 1) to examine

the affection from the zone. In Fig. 2, we can clearly see the real-time PSHA results corresponding to Fig. 1. When we remove the zone (Fig. 1b), the seismic intensity just slightly decreases in the southeast coast area and southern Taiwan, and the high intensity area in southeast ocean disappears (Fig. 2b). Moreover, the seismic intensity estimated by the GMPE rapidly attenuates from the zone and there is no influence in the northeast area in the estimation of seismic intensity. Thus, although the zone contributes the affection of seismic intensity, it is not significant on land or even negligible.

3. Why do you use only the ROC diagram and have not used others methods that provide important information about the performance of your model?

The ROC test already discusses and verifies the relationship between the space distribution of the forecasting probabilities and number of earthquake events. Under the concept of dichotomy, it is intuitive to shows what the ratio of target earthquakes are hit under the certain percentage of the area of probability distribution. In the calculation, the relationship between the spatial location of the earthquake and the probability distribution is examined. The increased ratio in y-axis represents the ratio of hit target earthquakes, and the shifting in x-axis represents the percentage of the area of probability distribution. Moreover, ROC test would give an absolute value from 0 to 1, not a relative evaluation, which is much intuitive and decisive to show the performance. Therefore, we chose it to test our results. On the other hand, there are still other testing methods presenting the performance. If our goal is to compare the performance between the forecast models, we should be under the same forecasting conditions and test methods to examine that. In our case, we focus on the concept and the calculating process of real-time PSHA so that we simply show that the forecasting results are good enough to be a probability function in the real-time PSHA calculation.

[Figure]

**Fig. 1.** Disassembled probability distribution. (a) Forecasting probability map of the Hualien earthquake from the PI. (b) Remove the zone (9 cells). (c) Only the zone (9 cells).

[Figure]

**Fig. 2.** Seismic intensity forecasting maps corresponding to Fig. 1. (a) Map of forecasted maximum seismic intensity for Hualien earthquake (b) The result of Fig. 1(b). (c) The result of Fig. 1(c).

---

## Author Response (AR1)

**Re: (nhess-2019-167) Real-time probabilistic seismic hazard assessment based on seismicity anomaly *by* Yu-Sheng Sun, Hsien-Chi Li, Ling-Yun Chang, Zheng-Kai Ye, and Chien-Chih Chen**

Dear Prof. Vallianatos,

Thank you for reviewing this paper. We have made the revision to our manuscript intensively and reply the comments from reviewers carefully for your further consideration on the publication in Natural Hazards and Earth System Sciences (*NHESS*).

The authors highly appreciate the support of publication in *NHESS* from the reviewers and their helpful suggestions as well. We have made substantive modifications according to their suggestions and the **English editing by Elsevier Language Editing Services**. We deeply appreciate their suggestions, which have made the manuscript become much better. The annotated responses to the reviewers' comments and the details about our changes in the revised version of our manuscript are made accordingly in the files.

Attached please also find the electronic files of the revised manuscript for your further consideration of publication in *NHESS*. In the revised version, all modifications were marked in red for your reference. Any problem raised please let me know. Thank you very much.

With Best Regards,
Yu-Sheng Sun

[Figure]

**Language Editing Services**

*Registered Office:*
Elsevier Ltd
The Boulevard, Langford Lane,
Kidlington, OX5 1GB, UK.
Registration No. 331566771

**To whom it may concern**

The paper "Real-time probabilistic seismic hazard assessment based on seismicity anomaly" by Yu-Sheng Sun, Hsien-Chi Li, Ling-Yun Chang, Zheng-Kai Ye, and Chien-Chih Chen was edited by Elsevier Language Editing Services.

Kind regards,

**Elsevier Webshop Support**

(This is a computer generated advice and does not require any signature)

**Response (in black) to the comments of Reviewer (in blue)**

Reviewer #1:
1. The description of the PI method.

We have added the description of the PI method. (Page 7-9, line 119-150)

[revised manuscript text omitted]

Reviewer #2:

1. The description of the PI method.

Thank you. Please refer to the reply to the first comment from Reviewer#1.
We have added the description of the PI method.

2. How do you explain that these values (the zone (9 cells) with the highest probability) do not affect the performance of your model?

We separated the probability distribution of Hualien earthquake (Fig. 1) to examine the affection from the zone. In Fig. 2, we can clearly see the real-time PSHA results corresponding to Fig. 1. When we remove the zone (Fig. 1b), the seismic intensity just slightly decreases in the southeast coast area and southern Taiwan, and the high intensity area in southeast ocean disappears (Fig. 2b). Moreover, the seismic intensity estimated by the GMPE rapidly attenuates from the zone and there is no influence in the northeast area in the estimation of seismic intensity. Thus, although the zone contributes the affection of seismic intensity, it is not significant on land or even negligible.

[Figure]

Figure 1. Disassembled probability distribution. (a) Forecasting probability map of the Hualien earthquake from the PI. (b) Remove the zone (9 cells). (c) Only the zone (9 cells).

[Figure]

Figure 2. Seismic intensity forecasting maps of real-time PSHA corresponding to Fig. 1. (a) Map of forecasted maximum seismic intensity for Hualien earthquake (b) The result of Fig. 1(b). (c) The result of Fig. 1(c).

3. Why do you use only the ROC diagram and have not used others methods that provide important information about the performance of your model?

The ROC test already discusses and verifies the relationship between the space distribution of the forecasting probabilities and number of earthquake events. Under the concept of dichotomy, it is intuitive to shows what the ratio of target earthquakes are hit under the certain percentage of the area of probability distribution. In the calculation, the relationship between the spatial location of the earthquake and the probability distribution is examined. The increased ratio in y-axis represents the ratio of hit target earthquakes, and the shifting in x-axis represents the percentage of the area of probability distribution. Moreover, ROC test would give an absolute value from 0 to 1, not a relative evaluation, which is much intuitive and decisive to show the performance. Therefore, we chose it to test our results. On the other hand, there are still other testing methods presenting the performance. If our goal is to compare the performance between the forecast models, we should be under the same forecasting conditions and test methods to examine that. In our case, we focus on the concept and the calculating process of real-time PSHA so that we simply show that the forecasting results are good enough to be a probability function in the real-time PSHA calculation.

[revised manuscript text omitted]